# Unexpected Radical Telomerisation of Vinylidene Fluoride with 2-Mercaptoethanol

**DOI:** 10.3390/molecules26113082

**Published:** 2021-05-21

**Authors:** Michel Duc, Bernard Boutevin, Bruno Ameduri

**Affiliations:** Université de Montpellier, CNRS, ENSCM, 34000 Montpellier, France; michel.duc64000@orange.fr (M.D.); bernard.boutevin@enscm.fr (B.B.)

**Keywords:** vinylidene fluoride, radical, telomerisation, mercaptan, chain transfer constant

## Abstract

The radical telomerisation of vinylidene fluoride (VDF) with 2-mercaptoethanol as chain transfer agent (CTA) was studied to synthesise fluorinated telomers which bear a hydroxy end-group, such as H(VDF)_n_S(CH_2_)_2_OH, under thermal (di-*tert*-butyl peroxide as the initiator) or photochemical initiations. A careful structural study of a typical H-VDF-S(CH_2_)_2_OH telomer was performed by ^1^H and ^19^F NMR spectroscopy. These analytical methods allowed us to explore the selective addition of the thiyl radical onto the hydrogenated side of VDF, and the telomer containing one VDF unit was obtained selectively. Surprisingly, for higher [VDF]_o_ initial concentrations, a monoadduct telomer was produced as well as PVDF homopolymer. This feature was related to the fast consumption of the CTA. The kinetics of radical telomerisation led to a quite high transfer constant of the CTA (40 at 140 °C) that evidences the formation of a monoadduct as the only telomer formed.

## 1. Introduction

Fluorinated polymers represent a unique class of materials that exhibit outstanding properties such as low surface energy; high thermostability; resistance to oxidation, chemicals (acids, oils and solvents) and ageing; low refractive index and water absorptivity; and excellent weatherability [1,2,3,4,5,6,7]. Hence, they can be involved in many applications such as coatings, optical fibres, high-performance elastomers, antifouling agents, piezoelectric devices, fuel cell membranes, separators, and polymer electrolytes for Li-ion batteries, cables and wires [1,2,3,4,5,6,7].

These (co)polymers are usually produced by radical (co)polymerisation. A valuable and practical approach is to consider the radical telomerisation of fluorinated olefins with a chain transfer agent (CTA or telogen), since that reaction is regarded as a suitable model of polymerisation [8]. Such a reaction can be initiated thermally, photochemically and by gamma rays, or in the presence of radical initiators or catalysts [3,8]. The molar mass of the resulting telomers greatly depends on the stoichiometry of the reactants, and most of all, on the chain transfer constant (C_T_) value: the lower the C_T_ value, the higher the molar mass of the resulting telomers [3]. This constant can be assessed from the ratio of the transfer rate to the propagation rate and various methods have been used to calculate it [3]. C_T_ is thus a key factor in determining the most probable reaction pathway. Although various articles report the C_T_ values of different chain transfer agents in the radical telomerisation of vinylidene fluoride (VDF) (Table 1), to the best of our knowledge, there is a lack of information for the assessment of the chain transfer constant of mercaptan. Such a CTA is rather well-used in thiol-ene reactions [9].

The radical telomerisation of VDF was comprehensively studied and reviewed 17 years ago [3]. Several kinetics of radical telomerisation of VDF were achieved from methanol [10], diethyl phosphonate [11], chloroform [12,13], bromotrichloromethane [12,13], carbon tetrachloride [12,13], BrCF_2_CFClBr [14], CF_3_I [15], C_6_F_13_I [16], HCF_2_CF_2_CH_2_I [16], xanthate [17], and iodine monochloride [18], and more recently, isopropanol, ethyl acetate and octyl acetate [19]. In addition, 1-iodoperfluoroalkanes [3,8] have also been successfully involved as efficient CTAs in iodine transfer polymerisation, either in supercritical CO_2_ [20] or under photochemical initiation catalysed by Mn_2_(CO)_10_ [21,22]. However, a few radical telomerisations of VDF have been reported with mercaptans [23,24], disulfides [25,26,27] or alkyl (or aryl) trifluoromethanethiosulfonates [28], while functional mercaptans have not been used and no transfer constants have been assessed. This gives rise to a further objective of this article, as well as the characterisation of the resulting telomers and unexpected side products.

## 2. Results

### 2.1. Radical Telomerisation of VDF with 2-Mercaptoethanol: Synthesis and Characterisation

#### 2.1.1. First Adducts

VDF being a gas, the first adducts of the telomerisation of VDF with 2-mercaptoethanol as CTA were synthesised in an autoclave using di-*tert*-butyl peroxide (tBuO)_2_, as the initiator (with an initial molar ratio to VDF: “C_0_” = [(tBuO)_2_)_0_/[VDF)]_0_ = 0.01) and acetonitrile as the solvent, at 140 °C (at this temperature, the half-life of di-t-butyl peroxide is ca. 1 h).

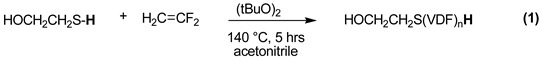


The initial mixture was composed of initial molar ratio R_0_ = [CTA]_0_/[VDF]_0_ = 1. The use of a solvent was required to improve the homogeneity of the liquid phase and the VDF dissolution in that medium, and acetonitrile was chosen because of its poor transfer activity in the presence of VDF [29]. Thus, at 140 °C, the initial pressure in the autoclave did not exceed 35 bar for 100 g of VDF (in a one litre-autoclave), in the presence of acetonitrile and then dropped to 11 bars. After the reaction, the overall VDF conversion was ca. 42%, according to the released fluorinated gas. After degassing, the reaction mixture presented as a slightly coloured liquid phase, which exhibited the characteristic nasty smell of the residual presence of thiol. After evaporation of the solvent, a distillation of the total product mixture was achieved to remove residual 2-mercaptoethanol and indicated that 50% of that chain transfer agent was converted. The product was then analysed by gas chromatography (GC), which evidenced the formation of one telomeric species (purity about 96% in response factors), further isolated by distillation. The monoadduct was characterised by ^1^H and ^19^F NMR spectroscopies (detailed in the experimental section). The ^1^H and ^19^F NMR spectrum (Figure 1a,b) for the distilled telomer highlights the exclusive Formulae (a):

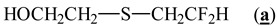


The ^1^H NMR (Figure 1b) and ^19^F (Figure 1a) NMR spectra allowed us to determine the structure of the monoadduct without any ambiguity, and therefore, the regioselective addition of HOC_2_H_4_S^•^ thiyl radical onto the methylene site of VDF. The addition of a single effective VDF unit was demonstrated by the presence of a triplet (^2^*J*_HF_ = 56.5 Hz) centred at 5.8 ppm, characteristic of a difluoromethyl -CF_2_-***H*** end-group in the ^1^H NMR spectrum of the product. This confirms previous studies that report the telomerisation of VDF with methanol [10], diethyl phosphonate [11] and chloroform [12]. The presence of the hydroxyl function, provided by the CTA, could also be identified by a triplet (^3^*J*_HH_ = 6.0 Hz), centred at 3.7 ppm, and a broad singlet that shifted with the concentration (or dilution) of the sample (hydroxyl-O***H***) [30]. The nature of the hydroxyl end-group was further confirmed by adding one drop of CCl_3_NCO in the NMR tube that induced a chemical shift of 1.0 ppm of the triplet, as a feature of the hydroxymethyl function and the vanishing of the broad singlet above. Finally, there was a distinction on the ^1^H NMR spectrum of the isolated product featuring two structures of methylene groups in the 2.6–3.0 ppm region. One of these groups was α-difluoromethyl, as evidenced by the nature of the resonance bands, namely, a triplet (^3^*J*_HF_ = 15.7 Hz) of triplets (^3^*J*_HH_ = 4.4 Hz). The other one was adjacent to another methylene group and resonated as a simple triplet (^3^*J*_HH_ = 6.0 Hz). Therefore, the general structure of the monoadduct of the telomerisation of VDF with 2-mercaptoethanol is in Formula (a) above.

The ^1^H NMR spectrum (Figure 1b) also reveals the absence of structure attributed to a CF_2_-CH_3_ end-group arising from the addition of CTA radicals onto the fluorinated carbon of VDF. Methyl end-groups would indeed be identified by a triplet (^3^*J*_HF_ = 19.0 Hz) centred around 1.8 ppm, as observed in the ^1^H NMR spectrum of the compound model CH_3_CF_2_CH_2_PVDF [10]. It can be concluded that the HOC_2_H_4_S^•^ radical was added selectively onto the less substituted carbon atom (CH_2_=) of VDF. In addition, that radical is electrophilic and thus reacts onto the nucleophilic methylene site. The regioselectivity of the addition was also confirmed by ^19^F NMR analysis of the monoadduct, which indicated a doublet (^2^*J*_FH_ = 56.5 Hz) of triplets (^3^*J*_FH_ = 15.2 Hz) centred at −113.9 ppm, characteristic of the fluorine atoms in CF_2_-H difluoromethyl end-group [10].

Infrared spectroscopy (Figure 2) [31] revealed the complete vanishing of the characteristic frequency of S-H bonds (ν = 2556 cm^−1^) initially present in the CTA, and the presence of specific bands assigned to difluoromethylene -CF_2_- groups at 1100–1200 cm^−1^.

The total product mixture, obtained after the telomerisation of VDF with equimolar quantities of 2-mercaptoethanol, contained the monoadduct (based on analysis GC) that represents more than 95% of synthesised telomers.

#### 2.1.2. Higher Molar Mass-Telomers

The telomers with higher degrees of telomerisation were synthesised in a 1 L-autoclave using the same initiator (C_0_ = [(tBuO)_2_]_0_/[VDF]_0_ = 0.01), a 10-fold excess of VDF (R_0_ = [CTA]_0_/[VDF]_0_ = 0.1) and acetonitrile as the solvent, at 140 °C. After 5 h reaction, the total of the reaction was purified and characterised by a marked staining (brown). According to Boutevin et al. [30], VDF-2-mercaptoethanol cotelomers of higher order are distinguished by very high boiling points and poor solubility in conventional organic solvents. To overcome the solubility issue, the obtained telomers were acetylated, making them soluble in polar solvents (such as acetone, DMF, DMSO, and NMP).

After evaporation of the solvent, the total product mixture was analysed by GC, the chromatogram of which (Figure 3) evidences the formation of a classical telomeric distribution with the first five produced adducts. The number-average degree of polymerisation, *DP_n_*, of these telomers was assessed (1.4), corresponding to an average molar mass, *M_n_*, of about 150, assuming that each adduct has the following formula: H(VDF)_n_S(CH_2_)_2_OH.

The ^19^F NMR spectrum (Figure 4) of the acetylated crude telomers displays main VDF units as highlighted by a doublet (^2^*J*_FH_ = 56.5 Hz), centred at −114.2 ppm, assigned to the fluorine atoms of difluoromethyl HC***F***_2_-end-group and an intense peak centred at −91.8 ppm corresponding to (-CH_2_-C**F_2_**-CH_2_-C**F**_2_-) [10,11,12,15,20,21,22] normal chaining in VDF−VDF dyads, while the reverse isomer exhibits a methyl end-group resulting from the reversed addition of a VDF unit as evidenced by the presence of a peaks centred at −94.7, −107.4, and −115.7 ppm [10,11,12,15,20,21,22]. These head-to-head defects of chaining, assessed by the corresponding integrals, were evaluated to 12%, which represents a high value (since usually PVDF contains 5–8% defects [4]).

The number-average degree of telomerisation, *DP_n_*, was also assessed by ^19^F NMR spectroscopy. The ratio of integral of the areas of difluoromethylene groups of both normal and reversed VDF–VDF sequences to that of difluoromethyl HC***F***_2_-end-group (Figure 3) gives *DP_n_* = 2.4. Surprisingly, this value does not match with the result obtained from the GC analysis.

The explanation for this discrepancy was provided by analysis of the telomers by size exclusion chromatography (SEC or GPC). The corresponding chromatogram (Figure 4) shows that the crude reaction product contained real telomers (including the first three adducts) and other species with a much higher average degree of polymerisation (ca. 5100 equivalent PMMA as the standards displayed with a negative polarity [32], Figure 5).

As expected, VDF oligomers of high molar masses were not detected by GC but could be observed in the ^19^F NMR spectra. The formation of these polymeric species could arise from the direct homopolymerisation of VDF in the presence of the remaining initiator after the total consumption of the CTA.

In fact, the possibility that these high molar mass species are “heavy” telomers, (i.e., with a high fluorine content) cannot be ruled out. It would explain the reversal polarity observed in the SEC chromatograms. Indeed, it was worth attempting to verify if polymeric species were still formed when the reaction was interrupted before the complete vanishing of the CTA. Thus, a telomerisation of VDF with the same CTA was carried out according to exactly the same protocol (including the same initial molar ratio R_0_ = 0.1), but within 30 min only.

After reaction and removing the solvent, the SEC chromatogram of the total product mixture (Figure 6) displayed the presence of telomers with low *DP_n_* only (thus highlighting the absence of any traces of polymers).

### 2.2. Kinetics of Radical Telomerisation of VDF with 2-Mercaptoethanol

The transfer efficiency of 2-mercaptoethanol in the presence of VDF is characterised by the transfer constant, C_T_^n^, that may be defined for each growing telomeric radical as the ratio of the transfer rate constant of the CTA, k_tr_^n^, to the rate constant of propagation of VDF, k_p_^n^, as follows:C_T_^n^ = k_tr_^n^/k_p_^n^

The evaluation of the kinetic constants of transfer of the first orders of such a telomerisation was performed using the method of David and Gosselain [33], which considers each telomer of i order as a CTA. Thus, the radical of order 1, HO(CH_2_)_2_S(CH_2_CF_2_)•, can react in two competing ways: (1) by transfer to CTA or (2) by propagation of VDF. These rate constants characterise reaction (1) of the formation of first order telomers from first order radicals, and reaction (2) of the formation of (n + 1) order telomer radicals from these same radicals, respectively, as follows:
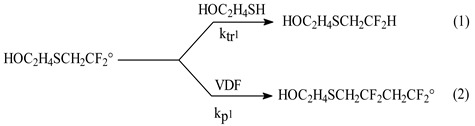


The produced telomers from step (1) are unable to participate in further radical reactions, especially as CTAs. Indeed, we have checked that HOCH_2_CH_2_SCH_2_CF_2_H produced by this telomerisation could not act as a new CTA when it was involved in the presence of organic initiator ((tBuO)_2_ or *tert*-butyl peroxypivalate) and VDF, in contrast to telomers/polymers of TFE [34] and VDF [16,20,21,35] with 1-iodo perfluoroalkanes, that were further used as original CTAs.

The transfer constant values of n order were obtained by assessing the slope of the straight lines representing the molar fractions of each telomer (obtained from the area of the corresponding peak in the gas chromatogram) versus R_0′_ for a given n value (Figure 7) [31].

We considered R_0_ as the ratio of the initial CTA concentration to the initial overall VDF concentration (i.e., the total VDF initial moles by volume of the liquid phase) [36].

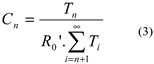



David and Gosselain’s law [33] enabled us to assess the transfer constant to 2-mercaptoethanol of 40 at 140 °C. This high value is compared to those of other chain transfer agents (Table 1) and is related to the dissociation energies of the S-H (307 kJ·mol^−1^), C-H (402 kJ·mol^−1^), P-H (322 kJ·mol^−1^), C_4_F_9_-I (225 kJ·mol^−1^), or I-Cl (211 kJ·mol^−1^) bonds [37]. That efficient transfer step can explain that remaining radicals, arising from unreacted initiator, are able to initiate the homopolymerisation of VDF, leading to PVDF. That observation was already noted in the radical telomerisation of styrene with mercapto-2-ammonium chloride that also generated polystyrene next to monoadduct telomer [38].

## 3. Materials and Methods

### 3.1. Materials

Vinylidene fluoride (VDF) was kindly supplied by Elf Atochem (now Arkema) (Pierre-Bénite, France), di-*tert*-butyl peroxide was generously offered by Akzo France (Chalons sur Marne), and 2-mercaptoethanol (99%) was purchased from Aldrich (Saint Quentin Fallavier, France). The reactants were used as received. Acetonitrile (Aldrich) was distilled from CaH_2_ and kept over activated 4 Å molecular sieves.

### 3.2. Reactions

To obtain a suitable amount of VDF telomers, the telomerisation was carried out in a 1 L Parr Hastelloy autoclave equipped with a manometer, a rupture disk, inner and outlet valves. An electronic device regulated and controlled both the stirring and heating of the autoclave. Prior to reaction, the autoclave was pressurised with 30 bar of nitrogen for 1 h to check for leaks. It was then conditioned for the reaction with several nitrogen/vacuum cycles (10^−2^ mbar) to remove any trace of oxygen. Di-*tert*-butyl peroxide, 2-mercaptoethanol, and acetonitrile were transferred into the autoclave under vacuum via a funnel tightly connected to the vessel. Then, after cooling in a liquid nitrogen/acetone mixture, VDF was introduced by double weighing (i.e., the difference of weight before and after filling the autoclave with the gases). The vessel was heated gradually up to 140 °C, and the evolution of pressure and temperature were recorded. The pressure increased up to ca. 25 bar, and after 4 h at the same temperature, it dropped to 11 bar. Reaction was stopped after desired times and the autoclave was cooled to room temperature and then placed in an ice bath. After purging the non-reacted monomer, the conversion of gaseous monomer was determined by double weighing. After the evaporation of acetonitrile, the total product mixture was fractionally distilled. The first fraction contained unreacted 2-mercaptoethanol for a [CTA]/[VDF] initial molar ratio lower than 1, while the other pure fractions were composed of various VDF telomeric adducts (mainly mono- and diadduct) and the residue was PVDF.

### 3.3. Apparatus and Analysis of Products

After reaction and evaporation of the solvent, the total product mixture was analysed by gas chromatography (GC) using a Delsi apparatus (model 330) equipped with an SE 30 column, 2 m, 1/8 in. (i.d.). The nitrogen pressure at the entrance to the column was maintained at 1 bar and the detector and injector temperatures were 240 and 235 °C, respectively. The temperature programme started from 50 °C and reached 220 °C at a heating rate of 10 °C/min. The GC apparatus was connected to a Hewlett Packard integrator (model 3390). The sensitivity of the apparatus and the resolution of the chromatogram limited the detection of telomers to the fifth adduct (Figure 2). The structure of the telomers was characterised by NMR spectroscopy at room temperature. The ^1^H, ^19^F, and ^13^C NMR spectra were recorded on Bruker AC-250 or Bruker AC-200 instruments using deuterated chloroform or acetone-d_6_, and tetramethyl silane or CFCl_3_, as the solvent and internal references, respectively. Coupling constants and chemical shifts are given in hertz (Hz) and in parts per million (ppm). The experimental conditions for recording ^1^H (or ^19^F) NMR spectra were the following: flip angle 90° (30°); acquisition time 4.5 s (0.7 s); pulse delay 2 s (5 s); 36 (200) scans and pulse width of 5 ms for ^19^F NMR. The letters s, d, t, q and m represent singlet, doublet, triplet, quartet and multiplet, respectively. The *IR spectra* were performed with a Nicolet spectrophotometer coupled to a compatible PC for data. acquisition. The position of the bands is given in cm^−1^, with an accuracy of ±2 cm^−1^. Average molar mass ($\bar{M}n$) and molar mass distribution ($\bar{M}w/\bar{M}n$, Đ) values were determined from chromatograms recorded by size exclusion chromatography (SEC or GPC) in DMF at 30 °C (flow rate = 0.8 mL·min^−1^) on apparatus equipped with an isocratic pump Spectra-Physics SP 8640 with an integrator and a computer unit for processing the chromatogram. The polymer samples were dissolved in DMF (2 mg·mL^−1^). All elution curves were calibrated with PMMA standards.

### 3.4. Typical Procedure for the Radical Telomerisation of VDF with 2-Mercaptoethanol: From an Initial Molar Ratio R_0_ = [CTA]_0_/[VDF]_0_ = 1

The VDF (100.0 g, 1.56 mol) reacted in a 1 L autoclave as described above with 2-mercaptoethanol (122.0 g, 1.56 mol) and acetonitrile (300.0 g, 7.31 mol) as the solvent. The telomerisation was initiated by di-*tert*-butyl peroxide (2.5 g, 0.0171 mol). The reactor was heated gradually up to 140 °C, and the evolution of pressure and temperature were recorded. After 5 h reaction (or 5 half-lives of the organic initiator), the telomerisation was stopped by quenching in an ice bath. After purging the non-reacted monomer, the conversion of gaseous monomer was determined by double weighing (the conversion rate of overall VDF was estimated at 42%). A fraction of the crude reaction product was removed for analysis (GC, SEC). After evaporation of acetonitrile, the total product mixture was fractionally distilled. The first fraction contained the residual thiol (b.p. = 157 °C or 65–66 °C/22 mm Hg), while the other pure fractions were composed of the various VDF telomeric adducts. The first adduct of telomerisation (m = 55.0 g) was isolated by vacuum distillation (b.p. = 101–103 °C/20 mmHg) with a purity of ca. 96% (from GC).

Monoadduct: 5,5-difluoro-3-thia-pentanol.

b.p. = 101–103 °C/20 mmHg, colourless liquid.

Results of ^1^H NMR (CDCl_3_) δ: 2.68 (t, ^3^J_HH_ = 6.0 Hz,-S-C**H_2_**-CH_2_OH, 2 H); 2.75 (td, ^3^J_H-F_ = 15.7 Hz; ^3^J_H-H_ = 4.4 Hz,-S-C***H*_2_**-CF_2_-H, 2 H); 3.39 (broad s, shifted with dilution -CH_2_O***H***, 1 H); 3.70 (t, ^3^J_HH_ = 6.0 Hz, shifted to 4.68 ppm with CCl_3_NCO, -C***H*_2_**OH, 2 H); 5.85 (tt, ^2^J_H-F_ = 56.5 Hz; ^3^J_H-H_ = 4.4 Hz, -CF_2_***H***, 1 H) ppm.

Results of ^19^F NMR (CDCl_3_) δ: −113.9 (dt, ^2^J_F-H_ = 56.5 Hz; ^3^J_F-H_ = 15.7 Hz, -C***F*_2_**H, 2 F).

### 3.5. Photochemical Telomerisation

The photochemical telomerisation was carried out by UV lamp irradiation (Philips HPK 125 W) of a sealed Pyrex tube (capacity: ~ 25 cc.) containing a mixture of VDF (16.2 g, 0.253 mole) and 2-mercaptoethanol (2.1 g, 0.027 mole), dissolved in acetonitrile (10.4 g, 0.25 mole) benzophenone (0.5 g, 2.70 × 10^−3^ mole) was used as a photoinitiator. After 12 h irradiation at room temperature, the reaction mixture was quenched into liquid nitrogen. Once the tube was opened, the mixture was degassed to remove the residual VDF (ca. 12 g). After evaporation of acetonitrile, the total product mixture was fractionally distilled. The first fraction contained the residual thiol, while the other pure fraction contained only the monoadduct, identified as the reaction product (checked by GC analysis and NMR spectroscopy).

## 4. Conclusions

The radical telomerisation of VDF with 2-mercaptoethanol led to ω-hydroxyl VDF-telomers. The stoichiometry of the reactants plays a crucial role in both the molar masses and the purity of the products. An excess of mercaptan led to quite low molar mass telomers only (especially HOCH_2_CH_2_SCH_2_CF_2_H monoadduct that resulted from the regioselective addition of thiyl radicals onto the less hindered site of VDF). When an excess of VDF was used, the monoadduct and slightly higher telomers (lower than hexadduct) were produced as well as PVDF homopolymer. Such an unexpected observation arose from the high transfer constant to thiol (C_T_ = 40 at 140 °C) that is quickly consumed, followed by the direct initiation of unreacted VDF from the remaining radical initiator. That efficient step is linked to the low dissociation energy of S-H bond in the mercaptan. For the first time, such a behaviour has been observed in radical telomerisation of fluorinated alkenes. The kinetic study further enabled completion of the series of C_T_ values of various CTA in telomerisation of VDF.

## Figures and Tables

**Figure 1 molecules-26-03082-f001:**
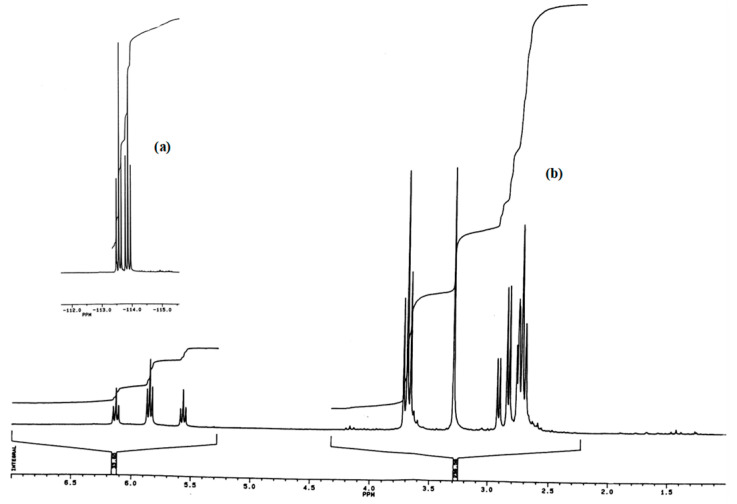
NMR spectra of HOCH_2_CH_2_SCH_2_CF_2_H monoadduct: (**a**) ^19^F NMR spectrum and (**b**) ^1^H NMR spectrum, recorded in CDCl_3_ at 20 °C.

**Figure 2 molecules-26-03082-f002:**
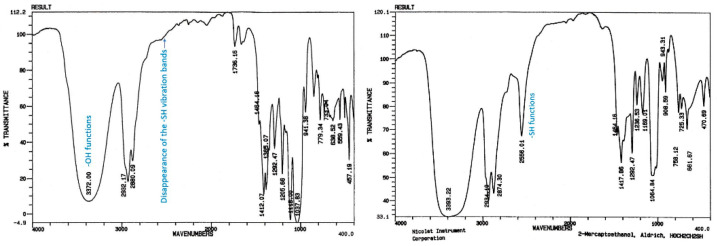
FTIR spectra of the total product mixture from the radical telomerisation of VDF with 2-mercaptoethanol (**left**) and of the 2-mercaptoethanol (**right**).

**Figure 3 molecules-26-03082-f003:**
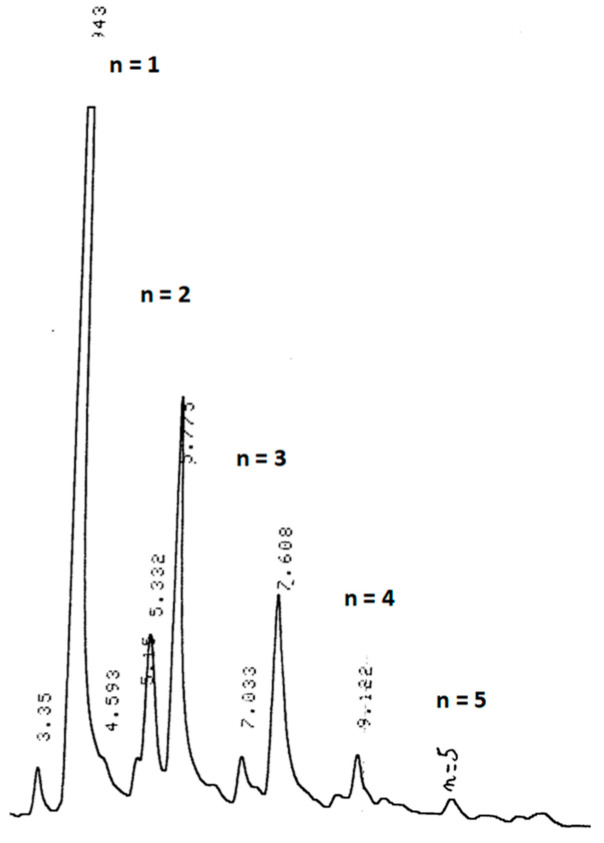
GC chromatogram of acetylated telomers mixture obtained from the telomerisation of VDF with 2-mercaptoethanol (initiated by di-*tert*-butyl peroxide, R_0_ = [CTA]_0_/[VDF]_0_ = 0.1, in acetonitrile as the solvent, at 140 °C, 5 h).

**Figure 4 molecules-26-03082-f004:**
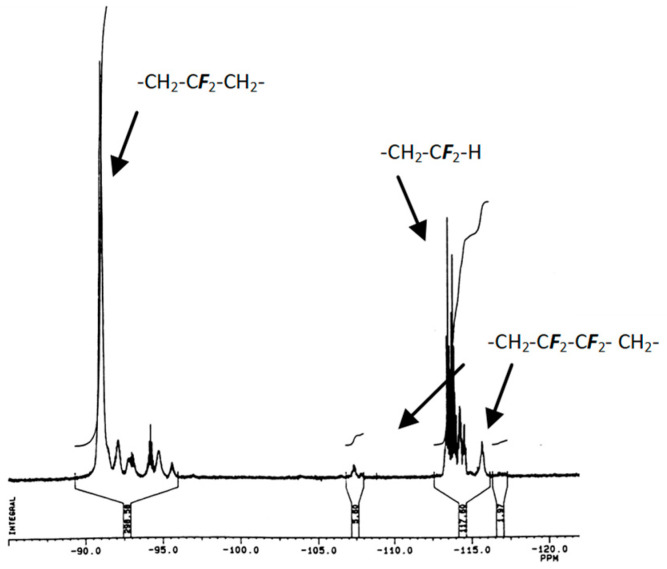
The ^19^F NMR spectrum of crude acetylated H(VDF)_n_S(CH_2_)_2_OCOCH_3_ telomers (from a radical telomerisation of a 10 folder excess-VDF with 2-mercaptoethanol) recorded in CDCl_3_ at 20 °C.

**Figure 5 molecules-26-03082-f005:**
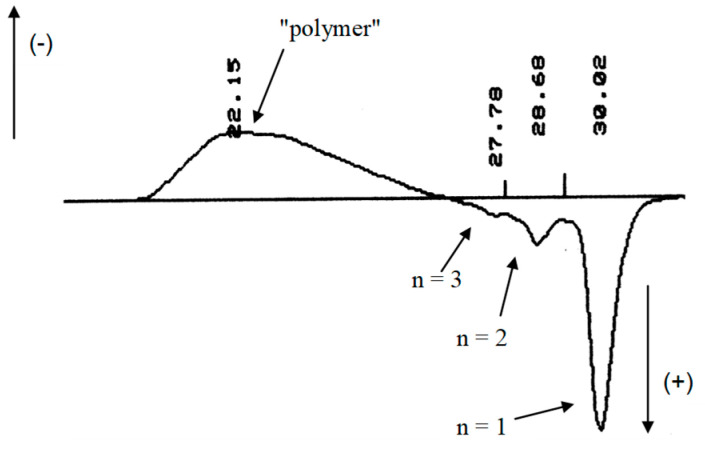
SEC chromatogram (in DMF, with a refractive index detector) of the acetylated telomers mixture obtained by telomerisation of VDF with 2-mercaptoethanol (using di-t-butyl peroxide as the initiator, R_0_ = [CTA]_0_/[VDF]_0_ = 0.1, acetonitrile as the solvent, at 140 °C, reaction times: 5 h).

**Figure 6 molecules-26-03082-f006:**
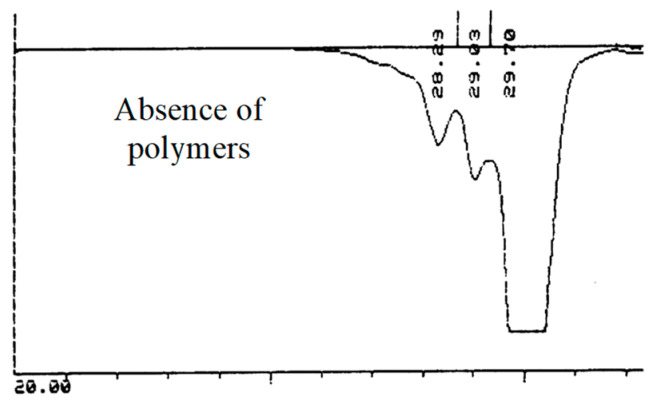
SEC chromatogram (in DMF with a refractive index detector) of the acetylated telomers mixture obtained by telomerisation of VDF with 2-mercaptoethanol (initiated by di-t-butyl peroxide, initial [CTA]_0_/[VDF]_0_ molar ratio of 0.1) and acetonitrile as the solvent, at 140 °C, reaction times: 30 min).

**Figure 7 molecules-26-03082-f007:**
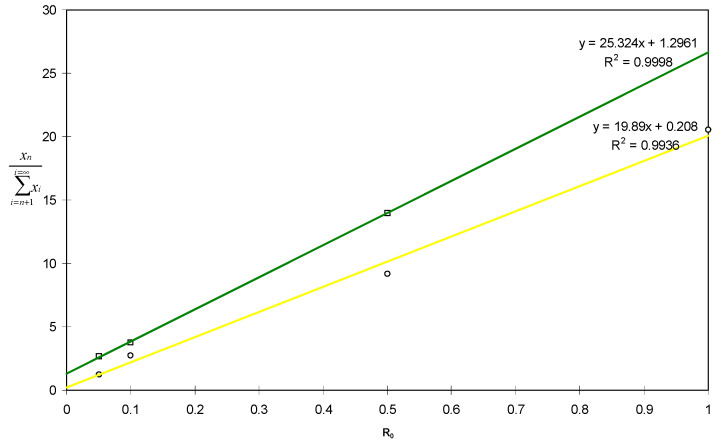
Molar fractions of each n telomer versus initial [CTA]_0_/[VDF]_0_ molar ratio (R_o_) for a given n value obtained from the telomerisation of VDF with 2-mercaptoethanol (initiated by di-*tert*-butyl peroxide at 140 °C, reaction times: 30 min).

**Table 1 molecules-26-03082-t001:** Summary of radical telomerisations of VDF with various chain transfer agents and their transfer constants.

Chain Transfer Agent	Transfer Constant (T/°C)	References
HOCH_3_	8 × 10^−3^ (140)	[10]
(EtO)_2_P(O)-H	0.34 (140)	[11]
Cl_3_C-H	0.06 (141)	[12,13]
Cl_3_C-Cl	0.25 (141)	[12,13]
BrCF_2_CFCl-Br	1.30 (75)	[14]
C_6_F_13_-I	7.4 (75)	[16]
HCF_2_CF_2_CH_2_I	0.3 (75)	[16]
Xanthate	49 (73)	[17]
Cl_3_C-Br	34 (141)	[12,13]
ICl	>40 (130)	[18]
RSH	40 (140)	This work

## Data Availability

Not applicable.

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
