# Peer review of "Unexpected Radical Telomerisation of Vinylidene Fluoride with 2-Mercaptoethanol"

_molecules, 2021, doi:10.3390/molecules26113082_

Round 1
Reviewer 1 Report
The manuscript by Ameduri and co-workers reported radical telomerisation of VDF with 2-mercaptoethanol, and analyzed by various analytical techniques, including NMR, GC-MS. The authors further highlight the influence of stoichiometry, and illustrated the kinetics. This work is of interest to readers and could be suitable for Molecules after consideration of the following points.
- The FT-IR studies of adducts should be studied.
- The English needs to be improved, for instance, line 33, subscript in Table 1.
Author Response
The manuscript by Ameduri and co-workers reported radical telomerisation of VDF with 2-mercaptoethanol, and analyzed by various analytical techniques, including NMR, GC-MS. The authors further highlight the influence of stoichiometry, and illustrated the kinetics. This work is of interest to readers and could be suitable for Molecules after consideration of the following points.
- The FT-IR studies of adducts should be studied.
Answer: The IR spectra of the product and of the 2-mercaptoethanl have been supplied in the revised version.
2. The English needs to be improved, for instance, line 33, subscript in Table 1.
Answer: A careful reading have been done to correct some typos.
Reviewer 2 Report
The presented work is an average scientific report. It add some information concerning the telomeraziation of vinylidene fluoride.
The research is well described. The litterature is appropriate, however, the formatting of litterature is a mess. Please revise 100% in journal format.
Photochemical iniatiation is not reported in the introduction part. The symbol° is used several times instead of the radical one fully filled circle.
See lines 88 and 112 for example
Figures have to be published into the highter possible quality and full spectra need to be placed in the SI part.
IR spectra of analysis described in pag 3 lines 118 - 120 are missed.
Equations 1, 2 and 3 have to be fully revised and placed in a more readable way
Author Response
The presented work is an average scientific report. It add some information concerning the telomerization of vinylidene fluoride.
The research is well described. The literature is appropriate, however, the formatting of litterature is a mess. Please revise 100% in journal format.
We appreciate the reviewers' comments and please find below our point by point answers.
The presented work is an average scientific report. It add some information concerning the telomeraziation of vinylidene fluoride.
The research is well described. The litterature is appropriate, however, the formatting of litterature is a mess. Please revise 100% in journal format.
answer: formatting has been imporoved and corrected following the instructions for authors.
Photochemical iniatiation is not reported in the introduction part. The symbol° is used several times instead of the radical one fully filled circle.
See lines 88 and 112 for example
answer: Though most studies (involving industrial ones are usually initiated thermally in presence of radical initiators, a few articles report the photochemical telomerization of VDF (we have cited 6 references on that specific initiation: 3, 7, 18, 21, 26 and 28)
Figures have to be published into the highter possible quality and full spectra need to be placed in the SI part.
answer we respectfully disagree with the reviewer and probably in the Word to pdf conversion, some troubles occured.
IR spectra of analysis described in pag 3 lines 118 - 120 are missed.
answer: the spectra of the total product mixture and mercaptan have been inserted.
Equations 1, 2 and 3 have to be fully revised and placed in a more readable way.
answer we respectfully disagree with the reviewer and probably in the Word to pdf conversion, some troubles occured.